# Insights into the Orchestration of Gene Transcription Regulators in *Helicobacter pylori*

**DOI:** 10.3390/ijms232213688

**Published:** 2022-11-08

**Authors:** Andrea Vannini, Davide Roncarati, Federico D’Agostino, Federico Antoniciello, Vincenzo Scarlato

**Affiliations:** Department of Pharmacy and Biotechnology (FaBiT), University of Bologna, 40126 Bologna, Italy

**Keywords:** *Helicobacter pylori*, environmental response, repressors of transcription, activators of transcription, two-component systems

## Abstract

Bacterial pathogens employ a general strategy to overcome host defenses by coordinating the virulence gene expression using dedicated regulatory systems that could raise intricate networks. During the last twenty years, many studies of *Helicobacter pylori*, a human pathogen responsible for various stomach diseases, have mainly focused on elucidating the mechanisms and functions of virulence factors. In parallel, numerous studies have focused on the molecular mechanisms that regulate gene transcription to attempt to understand the physiological changes of the bacterium during infection and adaptation to the environmental conditions it encounters. The number of regulatory proteins deduced from the genome sequence analyses responsible for the correct orchestration of gene transcription appears limited to 14 regulators and three sigma factors. Furthermore, evidence is accumulating for new and complex circuits regulating gene transcription and *H. pylori* virulence. Here, we focus on the molecular mechanisms used by *H. pylori* to control gene transcription as a function of the principal environmental changes.

## 1. Introduction

*Helicobacter pylori* is a human bacterial pathogen responsible for chronic active gastritis [1], gastric and duodenal ulcer diseases [2], and is associated with the development of gastric carcinoma [3]. Almost half of the world’s population carries an *H. pylori* infection, and disease outcome depends on many factors, including bacterial genotype, host physiology, and diet. The primary factors associated with pathogenesis include the urease enzyme [4], the flagellins [5], the vacuolating toxin VacA [6,7], and the cytotoxin-associated protein CagA [8,9]. As with other pathogens, the ability to acquire metal ions such as iron (Fe^2+^) [10,11] and nickel (Ni^2+^) [12,13,14] also seems to contribute to the virulence of *H. pylori*. Moreover, the heat-shock response plays a fundamental role during infection, as it allows cells to adapt to hostile environmental conditions and survive during stress.

Since the first *H. pylori* genome, strain 26695, was sequenced in 1997 [15], over 2000 isolates have been sequenced and annotated [16], and a representative reference genome reports a median chromosome of 1.63254 Mb in length coding for a median of 1440 proteins. Genome analysis revealed an abundance of polymorphic genetic elements, many of which reside in factors that may be associated with virulence [17].

Despite the enormous amount of information deduced from the genomes, our knowledge of many aspects of the molecular mechanisms regulating the physiology and metabolism of *H. pylori* is still limited. Most of the efforts focused on the bacterial and pathological processes of direct clinical relevance. In contrast, studies of the basic molecular mechanisms and the regulatory genes involved in the physiology and expression of virulence genes are still limited, and more efforts are required for better elucidation.

This review focuses on advances in molecular genetics studies of transcriptional functions, the mechanisms exerted by the regulators, and their interplays.

## 2. Genome and Regulatory Functions

In 1997, Tomb and collaborators published the genome sequence of the *H. pylori* strain 26695 [15]. The second genome sequenced was that of *H. pylori* strain J99 [18]. Although *H. pylori* were expected to exhibit a significant degree of genomic and allelic diversity, the overall genomic organization, gene order, and predicted gene products of the two strains were remarkably similar [18]. Moreover, other *H. pylori* strains have been sequenced, including strains G27 [19], N6 [20], and others.

The analysis of the *H. pylori* genome revealed that the genes encoding the basic transcriptional mechanism are very similar to those of other Gram-negative bacteria. Unlike most prokaryotes, the *rpoB* and *rpoC* genes encoding the β and β’ subunits of *H. pylori* RNA polymerase are fused into a single polypeptide. This peculiarity of gene fusion has also been observed in the closely related bacterium *H. felis* and the *Wolinella* species [21,22].

Overall, *H. pylori* codes for only three sigma factors, σ^80^ (RpoD), σ^54^ (RpoN), and σ^28^ (FliA) (Table 1), and it does not possess homologs of the stationary phase sigma factor (σ^s^) nor the heat-shock sigma factor (σ^32^). The absence of these factors suggested that *H. pylori* has a different stress response mechanism from other bacteria. Moreover, its genome contains four histidine kinases with their cognate response regulators and two orphan response regulators [23]. Furthermore, there are only a few other transcriptional regulators in *H. pylori*, which resemble repressors from other systems (Table 1). The low abundance of regulators might reflect the adaptation of *H. pylori* to its very restricted niche in the mucus layer of the human stomach, and the lack of competition from other micro-organisms [24].

## 3. The Two-Component Systems of *H. pylori*

Two-component systems (TCSs) are widespread signal transduction devices that couple the perception of external stimuli with an adaptive intracellular response. These systems are generally composed of two proteins, a histidine kinase (HK) and a response regulator (RR), both of which are characterized by different functional domains. Sensing a chemical or physical stimulus at the N-terminal input domain, the HK catalyzes the autophosphorylation of a conserved His residue on its C-terminal effector domain. The phosphate group is then transferred to a highly conserved Asp residue on the N-terminal receiver domain of the RR, leading to its activation [42]. Currently, different phosphotransmission schemes are known [43], enlarging the possible RR activation pathways. In most cases, RRs are characterized by C-terminal effector DNA-binding domains and act as transcriptional regulators [44]. Other types of RRs show RNA-, protein- or ligand-binding domains or are proteins with enzymatic activity, thereby broadening their spectrum of action to a post-transcriptional and post-translational level [44,45,46,47].

The *H. pylori* genome has evolved to better survive in the restricted colonization niche (i.e., the gastrointestinal tract), where it appears to be the dominant bacterial species and, therefore, with almost an absence of competition from other bacteria. Accordingly, efficient signaling networks have evolved to adequately express the factors necessary for survival in this harsh niche [48,49]. Compared to other bacteria, *H. pylori* encodes only four TCSs: FlgRS, ArsRS, CrdRS, and CheYA [15]. Two *H. pylori* histidine kinases (CrdS, ArsS) are likely transmembrane sensor proteins with a periplasmic input and a cytoplasmic transmitter domain. The atypical NtrB-like HK FlgS has revealed a cytoplasmic localization instead. Based on sequence similarities in their output domains, CrdR and ArsR RRs are classified as members of the OmpR subfamily, but FlgR is classified as an NtrC-like protein [23,50].

### 3.1. FlgRS Control the Transcription of Flagellar Genes

*H. pylori* flagellar genes are organized in a hierarchal regulon depending on the sigma factor required for their transcription, as σ^80^ (RpoD), σ^54^ (RpoN), and σ^28^ (FliA) govern the expression of early (class 1), intermediate (class 2 and class 3) and late (class 3) flagellar genes, respectively [27]. Due to its role in intermediate flagellar gene regulation, the TCS FlgRS has been shown to constitute an essential factor for *H. pylori* motility, hence, to host colonization [51]. Following stimulation, FlgS transfers its phosphate group to the cognate FlgR that directly interacts with RpoN, allowing the expression of its regulon. RpoN binding and subsequent activation were shown to occur in an enhancer-independent way, as FlgR lacks the typical C-terminal DNA-binding domain generally found in most σ^54^-dependent activators [26]. Interestingly, in mutants lacking FlgS or FlgR, an increase in *flaA* level was observed, a class 3 flagellar gene whose transcription is under the FliA control, suggesting a possible inverse regulation [27].

However, an intracellular signal leading to FlgS autophosphorylation and consequent FlgR activation remains to be elucidated [52]. Tsang and coworkers [53,54,55,56] hypothesized that different components of the flagellar export apparatus could coordinately operate to stimulate the autokinase activity of FlgS, as deletion of several of those proteins (FlhA, FliF, FliM, FliY, FliH, FliE, FlgB, FlgC) impaired the RpoN-mediated regulon. In the model they proposed, the correct assembly of the basal body and the Type 3 secretion system proteins of the flagellar apparatus generates a regulatory checkpoint for the on/off switch of FlgRS TCS. Moreover, they also demonstrated that the 25 N-terminus cytosolic residues of the export apparatus protein FlhA interact in vitro with FlgS via its C-terminal kinase domain. However, no increase in FlgS phosphorylation levels was observed in vitro in the presence of ATP [56]. Similar results were obtained in the close relative *Campylobacter jejuni*, where it was observed that the MS ring protein FliF and C ring protein FliG are necessary to activate FlgSR- and RpoN-dependent flagellar expression [57].

FlgS was further proposed to be a soluble pH sensor. In response to acidic (4.5) and strong acidic (2.5) cytoplasmic pH, FlgS mediates the transcription of an FlgR-independent regulon of over a hundred targets, comprising acid acclimation genes (*ureA*, *ureB*, *ureI*, *ureF*, *ansB*, *rocF*, *amiE*). FlgS is unable to bind directly to the promoters of the genes belonging to this large pH-responsive regulon. Furthermore, FlgR is not required for the regulation, and the factors that mediate the transcriptional response are still unknown. Most FlgR-independent FlgS-regulated genes (82%) are not present in the ArsS regulon, suggesting that FlgS and ArsS adopt different signaling pathways in response to gastric acidic pH. In addition, FlgS was found to be essential for *H. pylori* survival at pH 2.5 [58].

### 3.2. ArsRS Regulates the Transcription of Genes Involved in Acid Resistance

It is commonly recognized that the acid-adaptive response of *H. pylori* is tightly controlled, with ArsRS TCS at the top of the list of acid-responsive transcriptional regulation [59,60]. The pH-responsive regulon of ArsRS includes genes involved in acid adaptation (*ureAB*, amidases), oxidative stress responses (*katA*, *sodB*), iron or nickel homeostasis transcriptional modulation (*fur*, *nikR*), genes encoding outer membrane proteins (including *sabA*, *alpA*, *alpB*, *hopD*, and *horA*), and three genes coding for acetone carboxylase subunits (*acxA*, *acxB*, and *acxC*) [61]. The environmental pH is detected directly by the N-terminal periplasmic domain of the HK ArsS, which autophosphorylates and transfers the phosphate group to its cognate RR ArsR. The TCS ArsRS appears to be negatively autoregulated, as an ArsR binding site was found downstream of the *arsR* transcriptional start site (TSS) [59,62]. The ArsR regulon is composed of two sets of target genes: (I) genes controlled by the unphosphorylated ArsR, at least one of which is essential for in vitro growth conditions, and (II) non-essential genes which are regulated by the phosphorylated ArsR in response to acidic pH. The observation that an ArsS-deficient mutant is unable to colonize mice further emphasizes the role of the ArsRS in the acid adaptation of *H. pylori* in the host [63]. When *H. pylori* is exposed to low pH, the transcription of *ureAB* and the *ureIEFGH* operon is induced. This transcriptional initiation is mediated mainly by the ArsRS two-component system, as the effect was largely reduced in an ArsS-deficient mutant [60,64]. It was also demonstrated that the pH- and Ni^2+^-dependent transcription of *ureA* is mediated by two independent regulatory mechanisms involving transcriptional activators (NikR and ArsR) that compete for interaction with partially overlapping binding sites (see NikR) [28].

In response to acidic pH and the presence of low amounts of urea substrate for urease, *H. pylori* has additional mechanisms for ammonia synthesis. Two aliphatic amidases, AmiE and AmiF, appear to play a key role in nitrogen metabolism. Both amidase genes are transcribed in response to acidic pH, a response predominantly controlled by the ArsRS regulatory network [65,66]. Moreover, a dual regulation of *acxABC* and *fabD-pfs* operons by the ArsRS TCS and the orphan response regulator HP1021 was observed, indicating a central role in the pathway that sees acetone as an alternative carbon source. In fact, mutagenesis on the *acxABC* operon was shown to hamper *H. pylori’s* capacity to colonize mice [67,68]. Recent data have shown that the ArsRS plays a crucial regulatory role even in the absence of a low-pH condition, suggesting that it might be responsive to other environmental stimuli besides low pH [61].

### 3.3. CrdRS Regulates the Transcription of Genes Important for Copper Homeostasis and Central Cellular Responses

As a cofactor of oxidases (such as *cbb3*-type cytochrome-c oxidase), copper is critical for respiration in *H. pylori*, as it promotes the bacterial colonization of mucosal surfaces. On the other hand, copper acts as a chemotactic factor that repels *H. pylori* motility [69,70]. Therefore, copper homeostasis plays a crucial role in adaptation to the gastric environment [63], making necessary a fine regulation of the *H. pylori* intracellular copper level [71]. *H. pylori* has developed sophisticated efflux techniques to avoid copper toxicity by upregulating several genes involved in copper homeostasis, including the *copAP* operon [71]. In the presence of copper, the sensor kinase CrdS phosphorylates the cognate regulator CrdR that promotes the transcription of *crdAB* and *czcAB*, resulting in the expression of a secreted copper resistance protein (CrdA) and a copper efflux complex (CrdB, CzcB, and CzcA) [29,72]. CrdB, CzcB, and CzcA usefully complete the copper resistance mediated by the metal export pump CopA (a P-type ATPase). *H. pylori* CrdS and CrdR mutants were not able to colonize mice, further supporting the hypothesis that the copper resistance is mainly guided by the TCS CrdRS [29]. As shown by Hung and coworkers [73], *H. pylori* 26695 *crdS*, but not *flgS* nor *arsS*, was significantly upregulated in the presence of NO (nitrosative) stress, suggesting that CrdRS governs the expression of the NO-dependent regulon. In addition, this TCS was seen to play a role in increasing iron uptake for bacterial proliferation by upregulation of the iron-scavenging systems. Thereby, CrdRS makes a critical contribution to survive the host’s innate immune response [73,74,75].

### 3.4. CheY1Y2A Two-Component System

A functional chemotactic response is an essential determinant for *H. pylori* swimming motility and hence host colonization [76]. The *H. pylori* signaling cascade responding to a chemotactic signal mostly resembles that of the two archetypes *E. coli* and *Salmonella enterica* serovar Typhimurium [77]. As chemotaxis regulation by the CheAY TCS does not involve transcription responses, this essential topic is beyond the scope of this review, but will be briefly summarized hereafter. More detailed descriptions are reported in recent reviews [78,79]. The fundamental proteins involved in signal transduction from chemoreceptors to the flagellar switch are CheAY2, which consist of the CheA HK fused with the CheY-homolog receiver domain CheY2, and the CheY1 RR. In the absence of the specific ligand of the chemoreceptor, the HK CheA actively phosphorylates itself, and then the phosphate is transferred to an aspartate residue of the CheY1 RR. In turn, CheY1-P directly interacts with the flagellar motor and induces a clockwise rotation of the flagella, inducing a random reorientation in the space of the bacteria (tumbling behavior). Conversely, the binding of the ligand to the chemoreceptor inhibits the activity of CheA. Hence CheY1 remains unphosphorylated, and its interaction with the flagellar motor induces a counterclockwise rotation, prompting the bacteria to swim straight. In vitro, CheA transfers its phosphate group to CheY1 and CheY2, and CheY1 can transfer the phosphate back to CheA. It has been proposed that CheY2 interferes with the phosphate flow between CheA and CheY1, functioning as a phosphate sink. Three CheV phosphatases (homolog to *E. coli* CheW) target CheA-P and dephosphorylate the protein before phosphotransfer to CheY, but the former activity is less efficient than the latter [30]. Moreover, in a study conducted by Terry and coworkers [80], the ORF HP0170 was identified as a CheZ homolog that specifically directs the dephosphorylation of CheY.

### 3.5. Orphan Regulators

The genome of *H. pylori* encodes two DNA-binding regulatory proteins, named HP1021 and HP1043, that are defined as “orphan” regulators. They have been classified as response regulators based on sequence similarity to members of this protein family, but no histidine kinase, which is able to phosphorylate response regulators HP1021 and HP1043, has been identified. Considering that the phosphorylation receiver domains of both regulators show profound deviations from the consensus sequence and the experimental evidence that both proteins can bind in vitro target DNA without phosphorylation, the current hypothesis is that HP1021 and HP1043 may exert their regulatory function in the absence of receiver phosphorylation [81,82].

HP1043 proved to be essential for bacterial growth. The *hp1043* gene cannot be inactivated by allelic exchange, nor the amount of protein modulated, supporting its role as a central regulator of *H. pylori* fundamental cellular processes [82,83]. Indeed, a recent study based on chromatin immunoprecipitation-sequencing approach (ChIP-seq) identified genome wide the HP1043 in vivo binding sites [31]. This analysis identified 37 highly reproducible binding sites, ~90% of which are located in promoter regions, consistent with HP1043 functioning as a canonical transcriptional regulator. Gene ontology analysis of the identified targets revealed that HP1043 exerts a pleiotropic function, directly regulating all the fundamental processes in the cell, including translation, transcription, replication, energy metabolism, and virulence. To date, little is known about the environmental signals that regulate HP1043 expression and activity. A first observation proved that *hp1043* transcription is growth-phase regulated, with the amount of transcript dropping in the late stationary phase [83]. A recent report shows a significant increase in *hp1043* transcription following exposure of stationary cultures to urea, low pH, metals, biofilm, and AGS gastric epithelial cells [84]. In addition, HP1043 might be indirectly involved in response to oxidative stress since treatment of growing cells with redox-active compounds determined a decrease in protein levels and a reduction in the transcripts of regulated genes [85].

The inactivation of the HP1021 orphan response regulator leads to severe growth defects, but it is not essential for *H. pylori* viability [81]. A microarray-based whole genome transcriptional profiling study demonstrated that HP1021 controls the expression of 79 genes involved in different cellular processes, including transcription, translation, metabolic pathways, and synthesis of cofactors [67]. HP1021 has also been proposed to take part in the regulation of *H. pylori* chromosome replication. It has been shown that HP1021 binds to the *oriC* site and precludes the interaction of the DnaA replication initiator protein with this site, thereby inhibiting DNA unwinding [86]. As for HP1043, the environmental signals that trigger HP1021 activity have remained elusive for many years. A recently published paper proposed that HP1021 is a redox switch protein controlling the oxidative stress response in *H. pylori* [87]. In detail, HP1021 possesses several cysteine residues that are sensitive, both in vitro and in vivo, to oxidation. Moreover, the DNA-binding activity of HP1021 depends on redox conditions, and it regulates the transcription of specific genes in an oxygen-dependent manner.

## 4. Heat-Shock Response

The regulation of heat-shock genes represents a crucial process that allows bacteria to respond and adapt to environmental cues. These regulatory systems ensure a tightly controlled expression of a group of genes coding for proteins playing a protective role. The regulation of heat-shock genes can be based on a positive or a negative strategy. Positive transcriptional regulation employs condition-specific alternative σ factors that come into action when the organism is exposed to an external stress, thus activating heat-shock genes’ transcription. Conversely, negative regulation is governed by dedicated repressors that bind specific operators and repress the heat-shock genes’ transcription during physiological growth conditions. Following exposure to a stress insult, these repressors detach from their operators, and transcriptional repression is relieved [88].

### HspR and HrcA Repressors

In *H. pylori*, the major heat-shock genes are grouped into three multicistronic operons (*cbpA-hspR-helicase*, *groES-groEL*, and *hrcA-grpE-dnaK*), whose transcription is induced following a variety of stress stimuli [33]. After a sudden temperature increase, these three heat-shock operons are promptly transcribed with a very rapid kinetic. Moreover, their transcription is stimulated by other stress signals, including high osmolarity, cytoplasmic accumulation of unfolded proteins, and acid shock, suggesting their crucial role in survival under low-pH conditions [89]. Besides genes encoding chaperone proteins, *H. pylori* heat-shock operons harbor the sequence coding for two repressors, which are homologs to *Bacillus subtilis* HrcA and *Streptomyces coelicolor* HspR. Following the observation that their inactivation leads to heat-shock operons’ strong derepression [32], several studies detailed HspR and HrcA’s direct role in the negative control of *H. pylori* heat-shock genes. DNase I footprinting experiments carried out with purified HspR protein showed direct binding of the repressor to all three operons’ promoters [32], while HrcA binds to *groES-groEL* and *hrcA-grpE-dnaK* operons’ promoters only [90]. According to these data, HspR autoregulates the transcription of its own operon and, in combination with HrcA, they co-repress *groE* and *dnaK* expression. The binding sites of both repressors include conserved sequences similar to the well-characterized HrcA and HspR recognition sequences, named CIRCE (for controlling inverted repeat of chaperone expression) and HAIR (for HspR associated inverted repeat), respectively [91]. Considering the position of the HspR binding site on *cbpA-hspR-helicase* around the site of transcription initiation, the mechanism of repression, in this case, is based on the occupancy of the RNA polymerase binding region. Conversely, the situation at co-regulated promoters appears to be much more complex. In fact, for these operons, the HspR binding sites map upstream of the core promoter, while HrcA binding takes place in the core promoter region, overlapping the promoter elements and the transcription start sites. In addition, although HrcA and HspR binding sites are separated by a few base pairs and both repressors are necessary for complete repression, in vitro protein-DNA interaction assays suggested that they bind their operators independently, without any direct interaction [91]. A possible reason could be that in vivo binding of HrcA to its target sequences is not efficient without the assistance of a functional HspR repressor. Interestingly, in the closely related bacterium *C. jejuni*, which displays an almost identical heat-shock regulatory circuit, HrcA and HspR bind flanking operators in a cooperative manner and directly interact [92]. A schematic representation of HrcA and HspR regulation is depicted in Figure 1.

Although structural data are missing for these regulators, some biochemical characterization of the *H. pylori* heat-shock repressors has been put forward in the last decades. HspR is a 15 kDa protein that has the capacity to form multimers and, in agreement with this observation, binds and protects extended DNA regions on heat-shock promoters [90]. It is also a stable protein with respect to temperature, and in vitro, its DNA-binding activity is unaffected by heat challenges. The partner repressor HrcA, instead, is an unstable protein with an experimentally determined melting temperature of around 40 °C [93]. This feature is the basis of the role of HrcA as the thermosensor of the heat-shock regulatory circuit. In other words, HrcA has the capacity to perceive temperature changes directly. A temperature rise of a few degrees above 37 °C leads to an irreversible unfolding of HrcA, thereby, provoking a massive loss of its DNA-binding capacity and a prompt derepression of chaperone gene transcription [93]. A similar behavior has been observed in just a few other repressors governing heat-shock gene regulation in diverse bacteria species, including *C. jejuni*, *Streptomyces albus*, *Yersinia*, *Salmonella*, and in the thermophilic *Geobacillus thermoglucosidasius* [94,95,96,97,98,99].

In addition to negative transcriptional regulation operated by HrcA and HspR repressors, the expression of heat-shock genes in *H. pylori* is also regulated at the posttranscriptional level. In analogy to what happens in several other microorganisms, this additional level of gene expression control involves the feedback action of two distinct chaperones [33]. Firstly, the GroEL-GroES chaperonin positively modulates the DNA binding affinity of HrcA for its CIRCE-like operators [91], a mechanism that resembles the GroE-HrcA functional interaction demonstrated in *Chlamydia trachomatis* and *B. subtilis*, among others [100,101]. Putting together experimental results obtained in *H. pylori* and similar systems working in other microorganisms, a possible model postulates that upon interaction with HrcA during normal growth conditions, GroE chaperonin would enhance HrcA DNA binding activity (Figure 1). Following the exposure to a stress insult, the cytoplasmic accumulation of unfolded proteins would sequester GroE, provoking a loss of HrcA DNA-binding capacity and heat-shock promoter derepression. Moreover, GroE chaperonin plays a crucial role also in assisting HrcA recovery after inactivation by a heat-stress. In fact, while the loss of HrcA DNA-binding capacity is irreversible in vitro, experimental data have provided evidence that in vivo temperature-inactivated HrcA can retrieve its activity after the temperature challenge [93]. The observation that in vitro GroE restores HrcA binding activity following heat-inactivation supports the idea that following the induction step, at least part of the denatured repressor is refolded by GroE and can participate, together with the newly synthesized HrcA, in transcriptional repression of the target promoters.

An additional posttranscriptional feedback modulation of the activity of a heat-shock regulator is operated by CbpA. This protein is encoded by the first gene of the *cbpA-hspR-helicase* operon and displays a high degree of homology (33% amino acid sequence identity) to the DnaJ-like co-chaperone of DnaK, named curved DNA binding protein A for its additional role as a nucleoid-associated protein involved in nucleoid structuring function [102]. While in other organisms, as for example *Mycobacterium tuberculosis* and *S. coelicolor*, the regulatory activity of HspR is modulated by the major chaperone DnaK [103,104], in *H. pylori* the DNA-binding activity of this repressor is hindered by the direct interaction with CbpA (Figure 1). Interestingly, this negative modulation takes place when the repressor is detached from the DNA and not when it is bound to its operator. Accordingly, in cells that overexpress CbpA, HspR-controlled genes appear deregulated [102]. Considering this negative effect exerted by CbpA on HspR, one hypothesis is that this regulation is required to fine-tune the shut-off response of the heat-shock genes in *H. pylori*. Moreover, it has been recently demonstrated that *H. pylori* CbpA is a multifunctional protein, able to stimulate the ATPase activity of the major chaperone DnaK and to bind DNA. In addition, this DNA-binding activity can be modulated upon the direct interaction with the heat-shock master repressor HspR, suggesting the existence of reciprocal crosstalk between these two proteins [105].

Besides the dual regulation of heat-shock operons described above, HrcA and HspR control the transcription of a large collection of genes encoding proteins involved in diverse cellular functions and not strictly related to the stress response. The identification of HrcA and HspR regulons derives from genome-wide gene expression studies employing microarray technology and modern RNA-sequencing approaches. In the first genome-wide study, whole transcriptome analysis using microarray hybridization of RNA extracted from wild type, *hrcA*, and *hspR* single and double mutant cells revealed the roles of these repressors in the regulation of 43 genes in either a positive or a negative fashion [91]. These genes encode proteins involved in several cellular functions such as metal homeostasis, stress response, and bacterial motility (Figure 1). In particular, *hspR* and *hrcA* knockout mutations resulted in a significant downregulation of several genes coding for proteins involved in the regulation and biosynthesis of the flagellar apparatus. In agreement with transcriptional data, phenotypic assays of the *hspR*- and *hrcA*-mutant strains evidenced a significant negative impact on bacterial motility on soft agar plates [91]. This observation parallels the situation described in the closely related bacterium *C. jejuni* [106]. However, considering that the regulators of flagellar gene expression are not affected by repressor mutation, further studies are needed to understand the connection between heat-shock regulation and flagellar gene expression.

To better characterize heat-shock gene regulation by HrcA and HspR, two recent works carried out global transcriptome analysis applying the RNA-sequencing technique [107,108]. These analyses confirmed the involvement of both repressors in the regulation of different crucial cellular functions, highlighting the involvement of both regulators in the control of several motility genes. Of note, by comparing the HrcA and HspR regulons, it appears that the overlap between the two sets of genes is restricted to a limited number of cases, mainly represented by the already known HrcA and HspR co-regulated genes [108]. To identify the genomic regions bound in vivo by HspR and, thereby, to discriminate between the genes directly controlled by HspR binding from those indirectly affected by *hspR* mutation, Pepe and collaborators [107] carried out chromatin immunoprecipitation assays with a specific anti-HspR antibody followed by deep sequencing (ChIP-sequencing). This analysis revealed that the number of HspR genomic binding sites identified in vivo is very limited. Indeed, ChIP-seq analysis evidenced only four HspR genomic binding sites, three of which are associated with the promoter regions of the heat-shock operons and one mapping inside the coding sequence of the *speA* gene [107]. In other words, HrcA and HspR repressors seem to control a restricted direct set of genes, binding to and controlling the transcription of a limited group of promoters, while their inactivation leads to the pleiotropic effects of the *H. pylori* transcriptome. The idea that HspR and HrcA directly govern restricted “core regulons” is in parallel with previous observations made in several distant bacterial species, including *S. coelicolor*, *Listeria monocytogenes*, and *M. tuberculosis* [106,109,110,111,112,113].

## 5. Metal Homeostasis

Transition metal ions are essential micronutrients of bacteria since they accomplish a plethora of essential cell functions. They are necessary co-factors for enzymes, contribute to the structure of molecules and macromolecules, and are employed in signaling. On the other hand, high levels of transition metals are toxic for bacteria since they substitute other ions bound to proteins, inhibiting the function of these macromolecules and promoting the formation of reactive oxygen species (ROS) that are harmful to the cell [114]. Hence, bacteria tightly control the homeostasis of metal ions, balancing their import, storage, and use [115].

*H. pylori* employs iron and nickel ions as co-factors of proteins that have essential functions for host colonization and survival. Specifically, iron is used in many redox enzymes with indispensable functions for the survival of the bacterium, such as detoxification (catalase, superoxide dismutase), respiration and energy production (cytochromes, hydrogenase), catabolic and anabolic reactions [116]. Nickel is another important transition metal employed in enzymes for acid acclimation (urease) and energy production (hydrogenase) [13,14]. Specifically, the nickel-dependent urease hydrolyzes urea, producing ammonia, and increasing the pH of the milieu to levels that allow the bacterium’s survival [12,117]. *H. pylori* orchestrates nickel and iron homeostasis by employing two transcriptional regulators: NikR and Fur.

### 5.1. NikR

The key regulator of nickel homeostasis in *H. pylori* is NikR, a nickel sensor and Ni^2+^-dependent transcription factor [14,35] whose orthologs are present in most bacterial and archeal species [118,119]. The metal-sensing mechanism is based on the ability of NikR to convert from apo (no nickel-bound) to holo (NikR-nickel complex) when the intracellular level of nickel ions rises over a certain threshold. At least, at neutral pH, it is widely accepted that only holo-NikR binds to the operator and regulates transcription providing the nickel-dependent responses, while apo-NikR is not active [120,121]. In *H. pylori*, NikR also intervenes in the transcriptional responses to acid shock and iron homeostasis, setting NikR as a master regulator of this bacterium [28,66,122,123]. The NikR minimal unit is a homodimer and consists of a ribbon-helix-helix DNA-binding domain (DBD) and a metal binding domain (MBD), with the latter containing both the binding sites for 2 Ni^2+^ and the element that allows a further dimerization of the homodimer. In solution, two homodimers assemble into a tetramer independently from the occupation of the metal binding sites, originating an MBD central core connected to two peripheral DBDs through apparently flexible inter-domain linkers. These linkers allow a certain freedom of movement of the two DBDs, that can assume alternative reciprocal orientations, most notably the open, closed-trans, and closed-cis conformations observed in the crystal structures [124,125,126]. The NikR operator consists of two hemi-operators with TRTTA and TAWTA consensus sequences, respectively, located at exactly two DNA helix turns, and a less conserved TY element in the midpoint of the 15nt spacer [127,128,129,130]. Each hemi-operator is bound by a DBD, and the simultaneous interaction of the two DBDs of the NikR tetramer with the corresponding DNA elements occurs when the protein is in *cis* conformation only. In the tetrameric MBD central core, each of the 4 Ni^2+^ is coordinated in a square planar geometry by one cysteine and three histidines, and metalation of the MBD bridges two subunits of the tetramer and induces rearrangements of the protein [124,125,126]. According to molecular modelling and in vitro assays, metalation of the MBD core alters the conformation and the flexibility of the inter-domain linkers, modifying the motility of the two DBDs and the ability of the tetramer to interconvert among the conformers. These fluctuations allow the transition of holo-NikR tetramer in cis conformation, which is stabilized and fixed by the sequential and cooperative binding of the DBDs to the two hemi-operators [131,132,133,134,135]. Hence, as schematized in Figure 2, the binding of holo-NikR to its operator involves a conformational selection and an induced fit recognition mechanism.

A wide range of NikR-DNA binding affinities have been observed, with a set of promoters targeted at high affinity and others at lower, in a two-tiered binding model that depends on the sequence of the NikR operator and the stoichiometry of NikR metalation, and that is related to the strength and the kinetic of the nickel-dependent regulation of the downstream genes [128,136]. Many studies contributed to the dissection of the holo-NikR regulon and defined a bona fide set of genes directly regulated by the TR. Other studies reported non-overlapping or contradictory results, probably because of differences in the experimental approach, in the type of stimulus (brief/intense vs. mild/prolonged), in the environmental conditions (media and supplements), and in the *H. pylori* strain employed [130].

NikR senses high levels of nickel ions that convert to holo-NikR and directly repress the expression of the nickel transporters *nixA*, *frpB2*, *frpB3*, *fecA3*, *ceuE*, *fecDE*, *exbB2-exbD2-tonB2*, *hopV**, and *hopW** (* = putative) to halt the inflow of nickel ions and prevent intoxication (Figure 2) [35,127,137,138,139,140,141,142,143]. Moreover, holo-NikR directly represses the expression of Fe/Ni-hydrogenase (*hydA*, *hydB*) [144] and some virulence factors against the host (*hcpC*) and putatively against other bacteria (*mccBC*, *dvnA*), indicating that the switch from nickel starvation to its availability likely reduces hydrogen-dependent metabolism and the virulence against the host and the co-commensal bacteria [130].

Conversely, in the same conditions, NikR coordinates the employment and storage of nickel ions. Specifically, holo-NikR directly induces the expression of the *ureAB* operon that codes for the nickel-employing urease metalloenzyme [35,137,145] and of the nickel-storage *hpn*, *hpn2*, and *groES/hspA* genes [144], although this regulation is strain-dependent and is likely also mediated by other regulatory factors (see Fur; [35,130,146]). Interestingly, holo-NikR directly modulates the expression of regulators of gene expression, including its transcripts, *fur*, and *arsR* TRs [127,147], and small regulatory RNAs (sRNAs) [130,148,149]. Negative autoregulation is typical of many TRs; it has the function of optimizing the concentration of the sensor/regulator protein for an effective response to the environmental signal and when the stimulus is prolonged in time, it is a way to reduce the sensitivity to the signal through the downmodulation of the sensor/regulator protein. Nickel-dependent repression of *fur* is mirrored by the Fur-dependent repression of *nikR*, resulting in the crosstalk of these two TRs [122,127]. The interconnection of nickel- and iron-dependent circuits originates a complex transition metal homeostatic system that is exploited by the bacterium to co-regulate shared targets, including the Ni/Fe hydrogenase, metal storage proteins (*hpn*, *hpn2*), the virulence arsenal regulated by either of these TRs, and to tackle with acidic acclimation that relies on the regulation by Fur, NikR and the ArsR/ArsS two-component system, which, in turn, is directly regulated by these two TRs [150].

Holo-NikR represses the expression of three bacterial non-coding sRNAs named Nrr1, Nrr2, and NikS/IsoB [130] and, at least for the well-characterized NikS/IsoB sRNA, the nickel-induced repression induces upregulation of its targets CagA and HofC [148,149]. Holo-NikR also directly represses the expression of genes apparently unrelated to nickel homeostasis (*phbA*, *scoB*, *atoE*) [130]. Expression analyses performed after mild but prolonged exposure to nickel ions likely pinpointed not only the direct regulon, but also the indirect effects mediated by TRs and sRNAs targeted by NikR [35]. The NikR-repressed promoters contain operators in positions that almost invariably overlap the -10 element and/or the TSS in compliance with the effect of transcriptional repression [35,127,130,131,137,138,139,141,143]. In contrast, in the few cases in which NikR functions as a transcriptional activator, the DNA-binding sites are mainly located far upstream of the -10 element, in positions that are compatible with the activation of transcription [127,130,137,144,147,150]. Notably, holo-NikR increases the binding affinity of the RNA polymerase to the *ureA* promoter, boosting *ureA* expression [151], while at the *arsR* promoter, it increases the expression of the downstream gene functioning as an anti-repressor of transcription (Figure 2). Specifically, Fur is the primary repressor of the *arsR* promoter, and multiple NikR and Fur operators are located within the promoter in a complex architecture of adjacent and overlapping elements. At high nickel levels, NikR competes with Fur for binding to the promoter, forces the detachment of the repressor Fur, and increases *arsR* expression [150]. Similar architectures are present at other promoters, including the *fur* and *nikR* promoters, to obtain complex metal-dependent responses [122,127]. Lastly, holo-NikR binds to intragenic elements with no apparent effects on transcriptional regulation [130]. The biological significance of the intragenic binding sites has not been elucidated yet. Still, they could be tentatively explained as “parking bays” for the TR that increase its concentration on proximal regulated promoters. Alternatively, NikR may function as a nucleoid-associated protein that binds to these loci for chromosome organization and may induce long-range regulation mechanisms [152] or may have no direct functional relevance.

NikR is also involved in acid acclimation since its nickel-dependent regulation changes at low pH values and because it controls the expression of *fur* and *arsR* that also regulate the acid-dependent responses [62,146,153]. In vitro, the affinity of holo-NikR to the operator is increased at pH 6 and, in the same conditions, also apo-NikR binds to the operators [146,154]. In vivo, the exposure to acidic stress boosts the NikR-dependent transcriptional regulation, enforcing both the repression of the metal importers and the induction of urease [60,61,62]. These effects are likely due to the observed pH-dependent allosteric regulation of the TR, a boost of the NikR expression, the increase in the bioavailability of nickel ions at lower pH levels, and the modifications of multiple regulatory circuits intersecting with the NikR regulon in response to acid acclimation [61,146,153]. The involvement of NikR in nickel and acid homeostatic circuits is biologically important because the enzymatic activity of urease is critical to buffer environmental acidification and allow the bacterium’s survival at low pH values. Still, the enzyme requires high amounts of nickel as a co-factor. Consequently, at low pH, NikR contributes to the acid response by modulating the nickel trafficking and boosting the expression of the nickel-consuming urease.

### 5.2. Fur

The ferric uptake regulator—Fur—is a ubiquitous TR primarily involved in sensing the intracellular concentration of Fe^2+^ and modulating gene expression in response to iron concentration to maintain the homeostasis of this metal [36,155]. Similar to NikR, Fur converts from apo to holo when the intracellular level of iron ions rises over a certain threshold and, for most bacterial species, holo-Fur actively represses its regulon employing iron ion as a co-repressor, while apo-Fur is non-functional. The Fur protein of *H. pylori*, and a few other bacteria, is an exception since both apo- and holo-Fur actively bind to specific DNA sequences on the promoters (Fur boxes) and regulate the transcription of the downstream genes, with each form recognizing distinct consensus sequences and regulating a specific group of genes [156,157]. Hence, in *H. pylori*, Fur acts as a transcriptional commutator, with holo-Fur that represses its regulon in iron-replete conditions using Fe^2+^ as a co-repressor similar to the other Fur orthologs, and apo-Fur regulates the apo-regulon in iron-depleted conditions and employs the iron ion as an inducer. A schematic representation of Fur regulation in *H. pylori* is shown in Figure 3.

Fur is also involved in other regulatory pathways, including the responses to acid, osmotic and oxidative stresses; and contributes to nickel homeostasis and to the modulation of motility and chemotaxis; hence, it is considered a master regulator of *H. pylori*. Structurally, the Fur monomer is formed by an N-terminal DBD with a winged helix-turn-helix motif (wHTH) that interacts with the DNA at the operator sequences [158], a C-terminal MBD/dimerization domain, and a hinge region that connects these two domains. The MBD/dimerization domain contains three metal binding sites: the structural S1 site binds to a zinc ion trough two paired CXXC motifs, and its metalation is critical for dimerization and functioning of Fur; the regulatory S2 site is the high-affinity Fe-sensing element, and it is located in the hinge region between the DBD and the MBD/dimerization domain; S3 is located near the dimerization domain, and it has been proposed that its metalation strengthens Fur–DNA affinity and contributes to apo-regulation [159,160,161]. In the absence of iron, apo-Fur forms a head-to-head homodimer through the interaction of the MBD/dimerization domains and, according to molecular modelling and in vitro assays, it binds to the DNA along a single interface, with the axis connecting the two DBDs of the dimer almost parallel to the DNA [156]. The apo-Fur operator has a TCATT-n10-TT consensus sequence, and according to the investigators, protein-DNA interaction occurs by the insertion of the loop between helices α1 and α2 plus some residues of helix α4 of the DBD in the major groove of the DNA [156]. In the presence of iron, metalation of the S2 site occurs, converting apo- to holo-Fur and inducing a conformation reorganization of the protein, which assembles in a homo-tetramer and higher order oligomers even in the absence of target DNA. The holo-Fur operator consensus sequence is a TAATAAT-n-ATTATTA inverted repeat, and in silico and in vitro studies suggest that the Fur-tetramer contacts the DNA with each dimer that binds to a hemi-operator, approaching at opposite faces of the DNA, with the axis connecting their DBDs positioned perpendicular to the DNA major axis. Specifically, the DBDs of Fur contact two hemi-operators and insert part of the helix α4 in the minor groove of the DNA [156]. Hence, Fur commutates iron-responsive transcriptional regulation by the discriminative readout of opposed DNA grooves. The apo-Fur regulation is not unique for *H. pylori*, but it is limited to a few bacteria that harbor an additional 9–10 amino acids located immediately upstream of the DBD [157,160,161].

Many investigators have dealt with the definition of the Fur regulon in response to iron deficiency or abundance and environmental signals, including oxidative, acidic and osmotic stresses, growth phase, and motility/chemotaxis. Omics approaches reported hundreds of genomic loci bound in vivo by Fur in both low and high levels of iron ions and long lists of Fur-regulated genes in response to the different environmental conditions, indicating that Fur is a pleiotropic regulator of *H. pylori* [162]. Lists of genes directly regulated by Fur are provided, excluding the constellation of additional targets not confirmed in other analyses, likely derived from the methodology, the experimental conditions, and the strain employed. In iron-limiting conditions (Figure 3), apo-Fur directly represses the expression of iron storage (*pfr*) and of genes coding for proteins that employ iron ions to produce energy or to neutralize oxidizing species, including hydrogenase (*hydA*, *hydB*, *hydC*, *hydD*, *hydE*), cytochromes (*cytc533c*), superoxide dismutase (*sodB*), and other redox enzymes (*oorD*, *oorA*, *oorB*, *oorC*) [156,163,164,165,166,167,168]. However, the regulation of *sodB* has been observed to be strain-dependent. Moreover, apo-Fur regulates the expression of the *cag*-PAI toxin (*cagA*) and other genes co-transcribed in the apo-operons with no obvious relation with iron homeostasis (*serB*, *futB*) [165,169,170]. Hence, in conditions of iron starvation, apo-Fur reduces the storage of iron and spares the consumption of the precious metal, along with the modulation of some virulence factors. Conversely, in iron-replete conditions (Figure 3), holo-Fur directly represses the expression of genes that code for iron importers (*fecA1*, *fecA2*, *frpB1*, *feoB*), for energy transduction systems to fuel these metal transporters (*exbB2*, *exbD2*, *tonB*), and for a nickel-storage system (*hpn2*) [37,141,162]. Hence, Fur shuts off iron uptake when the intracellular levels of the metal are too high. Holo-Fur represses the ammonia-producing aliphatic amidase (*amiE*) and, maybe, also the urease (*ureA*), although for the latter gene, only an indirect Fur-mediated regulation has been reported [171,172,173]. Moreover, holo-Fur represses the expression of virulence factors (*ggt*, *hofC*, *putA*), of biosynthetic proteins for vitamin B2 and B6 (*ribBA*, *pdxA*, *pdxJ*), and of other genes apparently unrelated to iron homeostasis (*speE*) [169,173].

Fur functions as an autoregulatory rheostat of transcription to fine-tune its expression in response to iron bioavailability [174]. Specifically, in the presence of high levels of iron ions, it represses its expression, likely to reduce the sensitivity of the homeostatic circuit to prolonged iron stimulation (see NikR). In contrast, Fur induces its expression during iron starvation, probably to enhance the sensitivity of the regulatory system to the metal and increase the apo-Fur repression [175]. Fur also contributes to the regulation of the *nikR* and *arsR* TRs. Since NikR also targets the promoters of these genes, the Fur-NikR cross-regulation enables the integration of different environmental signals and the generation of complex metal-dependent responses (see NikR; [28,150]). Interestingly, the NikR-Fur crosstalk is not limited to the reciprocal metal-dependent regulation but also includes the ability of Fur to sense and partially transduce the environmental signal of nickel excess in vitro and in vivo [127]. Both apo-Fur and holo-Fur function mostly as repressors of transcription, while examples of positive Fur-dependent regulation have been reported, although the mechanism of Fur-dependent transcriptional activation has not been elucidated [162,169].

Many Fur-regulated promoters harbor multiple apo- and/or holo-operators positioned in adjacent and overlapping loci, originating complex regulatory architectures that fine-tune the expression of the downstream genes. For example, the apo-Fur regulated *pfr* promoter contains three apo-operators to strongly repress the gene expression in iron-limiting conditions [164,168], while the holo-Fur-repressed *frpB1* and *fecA2* promoters harbor two holo-operators, the *arsR* promoter has three operators, to highly repress transcription in iron-replete conditions [141,150,164]. At least at the *arsR* promoter, adjacent holo-operators mediate the oligomerization of holo-Fur, inducing DNA compaction and encasing the TSS in a repressive macromolecular complex; conversely, multiple apo-operators are unable to induce the oligomerization of apo-Fur [150]. Other promoters contain multi-operators of apo- and holo-elements to obtain a complex transcriptional modulation: at the *fur* promoter, proximal holo-operators mediate iron-dependent repression, while distant apo-operator functions as an anti-repressor of Fur itself to enhance transcription in the condition of iron starvation [176]. At the Fur-NikR co-regulated *nikR* and *arsR* promoters, the two TRs compete for the binding of overlapping but distinct operators [127,150]. Fur binds to its operators with different affinities, hence the kinetic and strength of Fur regulation depend on the architecture of the promoter, on the affinity Fur-DNA, and on Fur metalation [127,169]. Many Fur binding sites have been mapped within intracistronic loci, suggesting that the TR can target specific operators downstream of the TSS and modulate transcription likely by steric hindrance or inducing long-range DNA compaction, or alternatively it can regulate additional intracistronic TSS or antisense transcripts [162,177].

Fur is intimately involved in the redox homeostasis of *H. pylori* since it regulates many redox enzymes and oxidant-detoxifying proteins that are necessary to neutralize the chemical warfare of the host immune system and since the production of reactive ROS molecules depends mostly on free intracellular iron ions [114]. Under oxidative stress, Fur is allosterically regulated to reduce the affinity to the apo-operators and alleviate the apo-repression, while holo-regulon is only marginally affected [178]. Hence, the genes of the apo-Fur regulon are also oxidation-inducible Fur regulatory targets. Since the apo-regulon enlists ROS detoxifying enzymes and the oxygen-labile oxidoreductase OorDABC, which both contain iron ions and are repressed in iron-limiting conditions, their de-repression in response to oxidative stress is likely a way to relieve the repression and employ the few iron ions to produce defenses from ROS or replace damaged enzymes. Fur also contributes to the acidic response since it controls the expression of the other regulators of acidic responses NikR and ArsRS, and because its regulon is differentially expressed in acidic conditions [66,127,150,166,179]. Specifically, at low pH, the holo-Fur regulon is significantly de-repressed except for *fur* mRNA that is strongly downregulated, while the regulation of the genes targeted by apo-Fur is only marginally deregulated. Interestingly, the holo-Fur repressed *amiE* and *ureA* genes result derepressed at low pH values, and this acid-induced response could be a mechanism to express high amounts of the ammonia-producing enzyme to buffer the acidic milieu and grant the survival of the bacterium [166,172]. Fur accumulates in the stationary phase [162], suggesting that this TR is involved in the growth-phase-dependent regulation of genes belonging to apo- and holo-Fur operons [141,171]. Moreover, in the presence of high salt concentration, the Fur-dependent transcriptional responses resemble those induced by iron-depleted conditions, suggesting an integration of the osmolarity environmental signal into the Fur regulation [180,181]. Finally, Fur regulates genes involved in motility and chemotaxis, including *flaB* [162,171], and Fur knock-out mutants are hypo-motile [182] or colonize different areas of the gastric epithelium with respect to the wild-type strain in an in vivo infection model, indicating defects in the chemotaxis system [183].

## 6. Conclusions

The understanding of *H. pylori’s* transcriptional response to environmental changes has improved over the past years, bringing intricate gene expression networks to light. For some *H. pylori* regulatory systems, we have understood the mechanisms underlying the transcriptional response (heat shock regulators HrcA and HspR, NikR, Fur). In others, we are still far from their understanding, and more efforts are needed. Harmonic orchestration of gene expression is a requirement for successful infection. Accordingly, a profound understanding of the mechanisms of action of a regulatory protein is an essential requirement for developing new antibacterial drugs for treating a bacterial infection in combination or as an alternative to antibiotic therapy (see [184] and references therein).

It is well-established that antibiotics efficiently combat infectious diseases. However, pathogens develop defense strategies against them as we use the drugs, resulting in less or no effectiveness. Therefore, new approaches to combat infectious diseases are needed. Accordingly, increasing efforts have been put forward in searching for new compounds able to inhibit the growth of pathogens by acting specifically on new molecular targets for novel drug discovery. As a result, many transcriptional regulators have been considered and studied as valid drug targets with promising results, including the *H. pylori* HP1043 regulatory protein [185,186]. Furthermore, the two-component systems (TCS) have received increasing attention [187]. These compounds should be capable of binding and specifically blocking the functioning of the transcriptional regulators necessary to express genes involved in bacterial pathogenesis. Indeed, bacterial TRs are ideal targets for new antibiotics because these proteins have no homologs in eukaryotes, and any drugs could be specific for the bacterium.

## Figures and Tables

**Figure 1 ijms-23-13688-f001:**
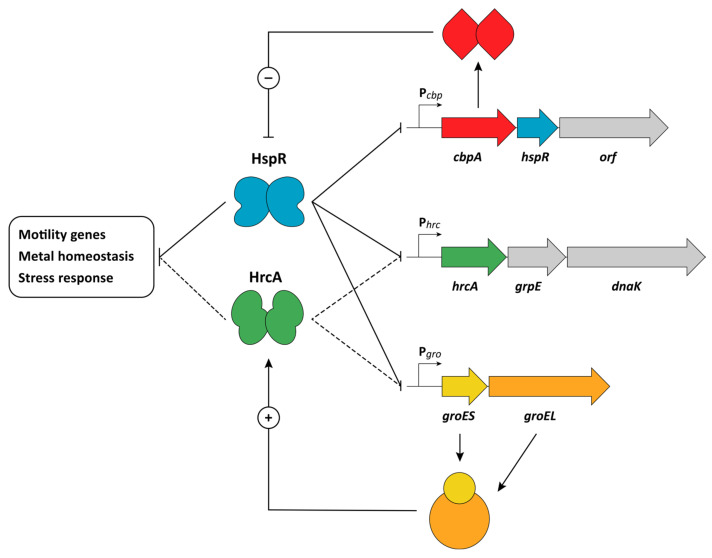
Schematic representation of HrcA (in green) and HspR (in blue) regulation in *H. pylori*. The colored and grey arrows indicate chaperone and regulatory genes, marked with their common names. P*_gro_* transcribes one bicistronic mRNA encoding the GroESL chaperonin machinery; P*_hrc_* transcribes a tricistronic mRNA encoding the repressor HrcA, the chaperone DnaK, and its cochaperone GrpE; and P*_cbp_* transcribes a tricistronic mRNA encoding the DnaJ homolog CbpA, the HspR repressor, and a putative DNA helicase (*orf*). CbpA (in red) and GroE (in yellow) contribute to the heat-shock response by regulating the activity of the HspR and HrcA regulators. Symbols: bent-arrows, direction of transcription; hammerhead, negative actions by the regulators to the indicated genes; “−” and “+” circled signs, negative and positive actions by the chaperone on the regulators, respectively; HspR and HrcA DNA-binding and transcriptional repression are represented by solid and dotted lines, respectively.

**Figure 2 ijms-23-13688-f002:**
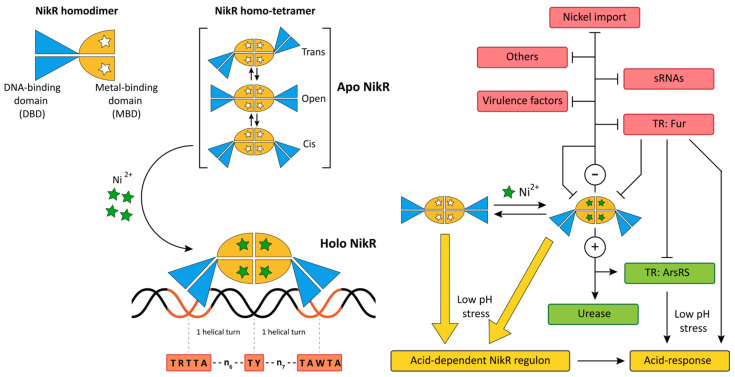
Schematic representation of NikR regulation. On the left part of the figure, a cartoon representation of NikR homodimer, homo-tetramer, and the Ni^2+^-mediated transition to holo-NikR, with the latter interacting with the reported operator consensus sequence. On the right part of the figure, the network connects NikR to its target set of genes. Symbols are as indicated in the legend in Figure 1.

**Figure 3 ijms-23-13688-f003:**
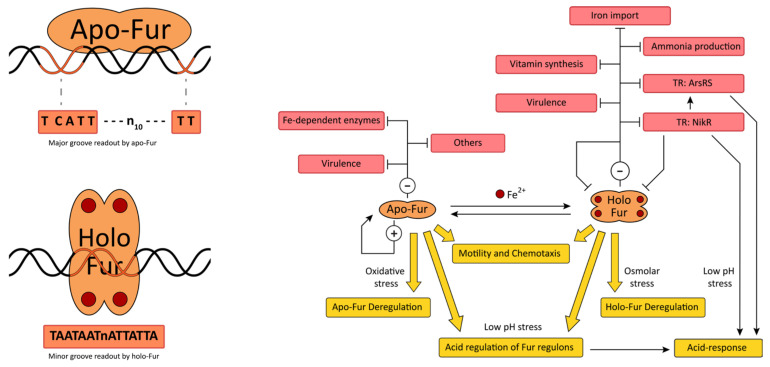
Schematic representation of Fur regulation in *H. pylori*. On the left part of the figure, a cartoon representation of the apo-Fur homodimer, holo-Fur homo-tetramer with the Fe^2+^-mediated (filled red circles) transition to holo-Fur, and their operator consensus sequences reported underneath, respectively. On the right part of the figure, the network connects apo- and holo-Fur to their target sets of genes. Symbols are as indicated in the legend in Figure 1.

**Table 1 ijms-23-13688-t001:** Transcriptional regulators predicted in the *H. pylori* genome.

Locus No. in Strain 26695	Putative Function (Gene)	References
**Sigma factors**		
HP0088	RNA polymerase sigma-80 factor (*rpoD*)	[25]
HP0714	RNA polymerase sigma-54 factor (*rpoN*)	[26]
HP1032	RNA polymerase sigma-28 factor (*fliA*)	[27]
**Histidine kinases and response regulators**		
HP0166/0165	Signal-transducing system (*arsRS*)	[23,28]
HP0703/0244	Signal-transducing system (*flgRS*)	[23,26]
HP1365/1364	Signal-transducing system (*crdRS*)	[23,29]
HP1067/0392	Signal-transducing system (*cheY1Y2A*)	[30]
HP1021	Response regulator	[23]
HP1043	Response regulator (*hp1043*, *hsrA*)	[23,31]
**Other regulators**		
HP1025	Chaperone gene repressor (*hspR*)	[32,33]
HP0111	Transcriptional repressor (*hrcA*)	[33,34]
HP1338	Nickel response transcriptional regulator (*nikR*)	[35]
HP1027	Ferric uptake regulator (*fur*)	[36,37]
HP1139	Putative repressor, (*soj*)	[38]
HP0222	Putative transcriptional regulator	[39]
HP0564	Putative transcriptional repressor	[40]
HP0835	Putative transcriptional regulator (*hup*)	[41]

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
