# Peer review of "Insights into the Orchestration of Gene Transcription Regulators in Helicobacter pylori"

_ijms, 2022, doi:10.3390/ijms232213688_

Round 1

Reviewer 1 Report

The review “Insightful of the orchestration of gene transcription regulators in
Helicobacter pylori” by Vannini et al.,
nicely summarizes our current understanding on the molecular mechanisms adopted by H. pylori to modulate its transcription profile in response to environmental changes. The authors clearly presented the main groups of H. pylori regulators and exhaustively analyzed their role in controlling gene expression. In my opinion, the review is comprehensive, up-to-date and undoubtedly complete in terms of references cited. Moreover, it is well written and includes three clear figures illustrating the mechanisms of action of some relevant regulators. Therefore, I strongly suggest the publication of this review in the present form on International Journal of Molecular Sciences.  

Author Response

We thank the reviewer for the positive comments.

Reviewer 2 Report

This is a well written, clear and comprehensive review of current knowledge of the transcriptional regulatory capacity of the gastric pathogen, H. pylori, by one of the leading groups in this field.  It brings together much of what is known in this regard and usefully highlights the complex inter-relationships between these activities.  Additionally, gaps in our knowledge are indicated where appropriate and thus this review will provide a fertile starting point for those seeking such information.  The review concludes on a promising note, speculating on the possibilities of targeting the transcriptional machinery from a therapeutic perspective.

Author Response

We thank the reviewer for the positive comments

Reviewer 3 Report

This is a major review that is badly needed in the field.  It is quite well-written and exhaustive.  It summarizes all, or at least nearly all the important recent work in the field.  I am recommending this for publication with only minor revisions.  I will summarize my suggested edits below.

The title, in its current form is a bit halting or awkward.  May I suggest an alternate that stays within your clever title construction?  In stead of "Insightful of...", I suggest "Insights into..."  Only a suggestion!!

The TCS section is well-done and covers the important material well.

Line 87-88;  This requires explanation.  I think you're refering to the specialized nature of Hp infection in the stomch only as opposed to more facultative or opportunist pathogens (S. aureus or P. aeruginosa).  This needs clarity as does the term "energy-saver" here.

lines 123-127. the 'FlgS mediate(s) the transcription of an FlgR-independent regulon...."need explanation of how this works.  Also, you list several genes as part of this regulon and then in the next sentence say that 82% of these gene are not part of the ArsS regulon.  Yet ureAB, rocF and aimE are members of the ArsRS regulon.  This needs clarity.

line 168;  Please explain/expand on the statement that "copper acts as a chemotactic factor against mucosal bacterial infection"

line 179;  there is a citation error here.  It was Hung et al. (73), not Loh & Cover (71) that showed CrdS responded to NO stress.  Also, that study used strain 26695, not J99.  Also, the role of CrdRS in chelating Fe is unclear.  This needs some explanation.

I know chemotaxis is outside the scope of this review, and transcriptional regulation is not a major feature of this HK/RR system (and this could be mentioned as a rationale to NOT describe the system extensively),  but the small paragraph  from line 187-197 doesn't do the topic justice.  I'd suggest minor elaboration AND the addition of citations to one or more relatively recent reviews for those interested.

line 275.  Re-thing the use of "contemplates" here.  I don't think it is the appropriate term.  Are you trying to sate that HspR binding "competes" with RNAP for binding to the promoter region?

Lines 395-412 ; great set up for the following sections!

line 425; "tetramerization element"  Needs explanation and it is not shown in Figure 2

lines 476-478;  "holo NikR directly represses the expression...." yet Figure 2 appears to suggest holo NikR induces ArsRS.  Please clarify.

line 510-511;  This line needs explanation/expansion.  Otherwise it seems only to be tacked on for no purpose.  It is a curious point!

Line 566;  the apo-Fur operator listed in text is slightly differenct than in FIgure 3 (TTT vs TT) - yes, this is a very minor point!!

Lines 578-580 appear to contradict a statement in lines 534-535

Lines 551=642;  This is an enormous paragraph.  Ut would be more easily read if it were broken up into three paragraphs.

In sum, this is a well-written, badly needed review.  I encourage its publication with minor changes.  It was a Herculean effort to pull all this together in this review and I thank you.  This will be invaluable to many in our field and it will be highly cited!

Author Response

This is a major review that is badly needed in the field.  It is quite well-written and exhaustive.  It summarizes all, or at least nearly all the important recent work in the field.  I am recommending this for publication with only minor revisions.  I will summarize my suggested edits below.

The title, in its current form is a bit halting or awkward.  May I suggest an alternate that stays within your clever title construction?  In stead of "Insightful of...", I suggest "Insights into..."  Only a suggestion!!

We accept the suggestion and thank the reviewer for it.

The TCS section is well-done and covers the important material well.

Thank you

Line 87-88;  This requires explanation.  I think you're refering to the specialized nature of Hp infection in the stomch only as opposed to more facultative or opportunist pathogens (S. aureus or P. aeruginosa).  This needs clarity as does the term "energy-saver" here.

We agreed and edited the sentence as follows (lines 87.91):

The H. pylori genome has evolved to better survive in the restricted colonization niche (i.e., the gastrointestinal tract), where it appears to be the dominant bacterial species and, therefore, with almost an absence of competition by other bacteria. Accordingly, efficient signaling networks have evolved to adequately express the factors necessary for survival in this harsh niche.

lines 123-127. the 'FlgS mediate(s) the transcription of an FlgR-independent regulon...."need explanation of how this works.  Also, you list several genes as part of this regulon and then in the next sentence say that 82% of these gene are not part of the ArsS regulon.  Yet ureAB, rocF and aimE are members of the ArsRS regulon.  This needs clarity.

The text has been revised and expanded as follows (lines 129-135):

FlgS is unable to bind directly to the promoters of the genes belonging to this large pH-responsive regulon. Furthermore, FlgR is not required for the regulation, and the factors that mediate the transcriptional response are still unknown. Most FlgR-independent FlgS-regulated genes (82%) are not present in the ArsS regulon, suggesting that FlgS and ArsS adopt different signaling pathways in response to gastric acidic pH. In addition, FlgS was found to be essential for H. pylori survival at pH 2.5

line 168;  Please explain/expand on the statement that "copper acts as a chemotactic factor against mucosal bacterial infection"

We attempted to clarify the concept by editing the sentence as follows:  copper acts as a chemotactic factor that repels H. pylori motility (line 176)

line 179;  there is a citation error here.  It was Hung et al. (73), not Loh & Cover (71) that showed CrdS responded to NO stress.  Also, that study used strain 26695, not J99.  Also, the role of CrdRS in chelating Fe is unclear.  This needs some explanation.

We apologize for citing the wrong reference and thank this reviewer for bringing it out. It has been corrected as suggested (line 188).

The role of CrdRS was clarified (lines 190-192): In addition, this TCS was seen to play a role in increasing iron uptake for bacterial proliferation by upregulation of the iron-scavenging systems.

I know chemotaxis is outside the scope of this review, and transcriptional regulation is not a major feature of this HK/RR system (and this could be mentioned as a rationale to NOT describe the system extensively),  but the small paragraph  from line 187-197 doesn't do the topic justice.  I'd suggest minor elaboration AND the addition of citations to one or more relatively recent reviews for those interested.

We accepted the suggestions and revised this chapter, accordingly (lines 199-218):

As chemotaxis regulation by the CheAY TCS does not involve transcription responses, this essential topic is beyond the scope of this review, but will be briefly summarized hereafter. More detailed descriptions are reported in recent reviews [78,79]. The fundamental proteins involved in signal transduction from chemoreceptors to the flagellar switch are CheAY2, which consist of the CheA HK fused with the CheY-homolog receiver domain CheY2, and the CheY1 RR. In the absence of the specific ligand of the chemoreceptor, the HK CheA actively phosphorylates itself, and then the phosphate is transferred to an aspartate residue of the CheY1 RR. In turn, CheY1-P directly interacts with the flagellar motor and induces a clockwise rotation of the flagella, inducing a random reorientation in the space of the bacteria (tumbling behavior). Conversely, the binding of the ligand to the chemoreceptor inhibits the activity of CheA. Hence CheY1 remains unphosphorylated, and its interaction with the flagellar motor induces a counterclockwise rotation, prompting the bacteria to swim straight. In vitro, CheA transfers its phosphate group to CheY1 and CheY2, and CheY1 can transfer the phosphate back to CheA. It has been proposed that CheY2 interferes with the phosphate flow between CheA and CheY1, functioning as a phosphate sink. Three CheV phosphatases (homolog to E. coli CheW) target CheA-P and dephosphorylate the protein before phosphotransfer to CheY , but the former activity is less efficient than the latter [29].

line 275.  Re-thing the use of "contemplates" here.  I don't think it is the appropriate term.  Are you trying to sate that HspR binding "competes" with RNAP for binding to the promoter region?

We agree with this observation and changed “contemplates” to “is based on” (line 298)

Lines 395-412 ; great set up for the following sections!

Thank you

line 425; "tetramerization element"  Needs explanation and it is not shown in Figure 2

We reworded the sentence as follows (lines 446-449): The NikR minimal unit is a homodimer and consists of a ribbon-helix-helix DNA-binding domain (DBD) and a metal binding domain (MBD), with the latter containing both the binding sites for 2 Ni2+ and the element that allows a further dimerization of the homodimer.

lines 476-478;  "holo NikR directly represses the expression...." yet Figure 2 appears to suggest holo NikR induces ArsRS.  Please clarify.

We clarified by changing the word “represses” to “modulates” (line 499).

line 510-511;  This line needs explanation/expansion.  Otherwise it seems only to be tacked on for no purpose.  It is a curious point!

We attempted to explain by adding the following text (line 535-540): The biological significance of the intragenic binding sites has not been elucidated yet. Still, they could be tentatively explained as ‘parking bays’ for the TR that increase its concentration on proximal regulated promoters. Alternatively, NikR may function as a nucleoid-associated protein that binds to these loci for chromosome organization and may induce long-range regulation mechanisms [152] or may have no direct functional relevance.

Line 566;  the apo-Fur operator listed in text is slightly differenct than in FIgure 3 (TTT vs TT) - yes, this is a very minor point!!

We are sorry for the typo. The extra T at the end of the consensus has been removed: TCATT-n10-TT (line 598)

Lines 578-580 appear to contradict a statement in lines 534-535

The apparent contradiction has been removed (lines 565-566): The Fur protein of H. pylori, and a few other bacteria, is an exception …

Lines 551=642;  This is an enormous paragraph.  Ut would be more easily read if it were broken up into three paragraphs.

We agree with this observation and broke the paragraph into three paragraphs.

In sum, this is a well-written, badly needed review.  I encourage its publication with minor changes.  It was a Herculean effort to pull all this together in this review and I thank you.  This will be invaluable to many in our field and it will be highly cited!

We thank you for the valuable comments and your appreciation.